# Transgenerational Paternal Inheritance of *TaCKX* GFMs Expression Patterns Indicate a Way to Select Wheat Lines with Better Parameters for Yield-Related Traits

**DOI:** 10.3390/ijms24098196

**Published:** 2023-05-03

**Authors:** Karolina Szala, Marta Dmochowska-Boguta, Joanna Bocian, Waclaw Orczyk, Anna Nadolska-Orczyk

**Affiliations:** Department of Functional Genomics, Plant Breeding and Acclimatization Institute—National Research Institute, Radzikow, 05-870 Blonie, Poland

**Keywords:** parental imprinting, transgenerational epigenetics, paternal inheritance, wheat, *TaCKX* expression, cytokinin, yield

## Abstract

Members of the *TaCKX* gene family (GFMs) encode the cytokinin oxygenase/dehydrogenase enzyme (CKX), which irreversibly degrades cytokinins in the organs of wheat plants; therefore, these genes perform a key role in the regulation of yield-related traits. The purpose of the investigation was to determine how expression patterns of these genes, together with the transcription factor-encoding gene *TaNAC2-5A*, and yield-related traits are inherited to apply this knowledge to speed up breeding processes. The traits were tested in 7 days after pollination (DAP) spikes and seedling roots of maternal and paternal parents and their F_2_ progeny. The expression levels of most of them and the yield were inherited in F_2_ from the paternal parent. Some pairs or groups of genes cooperated, and some showed opposite functions. Models of up- or down-regulation of *TaCKX* GFMs and *TaNAC2-5A* in low-yielding maternal plants crossed with higher-yielding paternal plants and their high-yielding F_2_ progeny reproduced gene expression and yield of the paternal parent. The correlation coefficients between *TaCKX* GFMs, *TaNAC2-5A*, and yield-related traits in high-yielding F_2_ progeny indicated which of these genes were specifically correlated with individual yield-related traits. The most common was expressed in 7 DAP spikes *TaCKX2.1*, which positively correlated with grain number, grain yield, spike number, and spike length, and seedling root mass. The expression levels of *TaCKX1* or *TaNAC2-5A* in the seedling roots were negatively correlated with these traits. In contrast, the thousand grain weight (TGW) was negatively regulated by *TaCKX2.2.2*, *TaCKX2.1*, and *TaCKX10* in 7 DAP spikes but positively correlated with *TaCKX10* and *TaNAC2-5A* in seedling roots. Transmission of *TaCKX* GFMs and *TaNAC2-5A* expression patterns and yield-related traits from parents to the F_2_ generation indicate their paternal imprinting. These newly shown data of nonmendelian epigenetic inheritance shed new light on crossing strategies to obtain a high-yielding F_2_ generation.

## 1. Introduction

Bread wheat (*Triticum aestivum*) is the most important cereal crop in the temperate climate and provides a staple food for more than a third of the world’s population [1]. It belongs to the Triticeae tribe, which also includes barley, rye, and triticale. Among the species, wheat has the largest and most complex hexaploid genome (2n = 6x = 42), which consists of the three homoeologous subgenomes A, B, and D. Each gene present as homologues A, B, and D could retain its original function or, as a result of independent evolution, develop heterogeneous expression, and/or one or two copies may be silenced or deleted [2,3].

Cytokinins (CKs) perform a basic role in the growth, development, and productivity of any plant species, including wheat [4]. Their content in developing spikes of wheat is correlated with grain yield, grain number and weight, TGW, chlorophyll content in flag leaves, and seedling root weight [5,6,7,8,9]. There are two types of cytokinins, isoprenoid and aromatic. The most important and widely occurring are isoprenoids, *cis*-zeatin (cZ), *trans*-zeatin (tZ), isopentenyl adenine (iP), and dihydrozeatin (DZ), and fewer aromatic forms, e.g., benzylaminopurine (BA). The content of active forms in plant tissues and organs depends on metabolic processes, such as biosynthesis, degradation, inactivation, and reactivation. Active forms and their ribosides can also be transported throughout the plant [10]. Enzymes for all metabolic processes are encoded by members of the gene family (GFMs) [11,12,13,14,15]. One of the most important processes is the irreversible degradation of cytokinins by *TaCKX* GFMs. There are 13 basic *TaCKX* GFMs, 11 of which have homoeologs in subgenomes A, B, and D. Two of them, *TaCKX2.2.2* and *TaCKX2.2.3*, are located only in the D subgenome [11]. The *CKX* GFMs encode the cytokinin oxidase/dehydrogenase enzyme. Their role in the regulation of yield-related traits in wheat [5,6,7,11] and other species were already shown [4,16,17,18]. Decreased expression of the *TaCKX1* or *TaCKX2* genes significantly influenced TGW, seed number, and chlorophyl content in flag leaves, and the effect was dependent on the silent gene and/or genotype [5,6,7]. The *TaCKX4* copy number affected grain yield and chlorophyl content in flag leaves [19]. Haplotype variants of *TaCKX6-D1* (actually *TaCKX2.2.1-3D*) were associated with TGW [20], and the allelic variant of *TaCKX6a02* (annotated as *TACKX2.1*) influenced grain size, filling rate, and weight [21]. In addition to yield, some *CKX* GFMs influence other pleiotropic traits, including root growth, nutrient accumulation, and abiotic stress responses [18,22,23]. These genes could also be regulated at the transcriptional level by transcription factors (TFs), especially those belonging to the NAC family; however, knowledge about their function is still very limited [24]. A promising NAC-encoding candidate with a role in yield-related traits is *TaNAC2-5A* [25,26]. As reported, overexpression of the gene delayed leaf senescence and increased nitrate uptake and concentration, root growth, and grain yield under field conditions. It is interesting to note that in a controlled environment, *TaNAC2-5A* was negatively correlated with the activity of the CKX enzyme in seedling roots and the number of tillers [8].

Crossbreeding and selection are basic steps in crop improvement, and the only potential limitation is too narrow genetic variability. Therefore, it is important to know how yield-related traits are inherited. Moreover, traits, including stably integrated transgenes and edited genes, are inherited according to Mendelian rules [27,28,29,30]. An exception to Mendel’s principles that encompass both groups of genes is epigenetic inheritance [31] or the polygenic nature of genomic architecture for the linked traits, which can be regulated by transcription factors [32,33,34] or other gene regulatory networks [35].

The pattern of gene expression could be considered a trait with its own type of inheritance.

This concept, reviewed by Yoo et al. [2], called parental expression additivity, is defined as the arithmetic average of the expression of the parental genes. Expression additivity of parental genes is observed in the offspring of the diploid species. The deviation of additivity called parental non-additive expression is mainly found in the offspring of polyploid species. A bias when the expression of the offspring is similar to that of one of the parents is called expression-level dominance. If total offspring expression is lower or higher than in both parents, the phenomenon is called transgressive expression, and when the contribution of the parental homeologs to the total gene expression is unequal, it is named homeolog expression bias. All this deviation from additivity can be explained as a result of different factors, such as the influence of one of the parental genomes, epigenetic regulation, balance of gene dosage, and *cis-* and/or *trans*-regulatory elements [2].

Expression-level dominance, which is of uniparental origin, is also called genomic imprinting [36,37]. This phenomenon of epigenetic origin is the result of the asymmetries of DNA and histone methylation between maternal and paternal plants. Male and female genotypes are multicellular in origin; therefore, primary gene imprinting can occur in egg cells, central cells, and sperm and, subsequently, in the triploid endosperm or, less frequently, in the diploid embryo [37]. Generally, conservation of imprinting is limited across other crops; however, these genes that show conserved imprinting in cereals showed positive selection and were suggested to perform a dose-dependent function in the regulation of seed development [38]. However, as recently documented by Rodrigues et al. [39], most genes imprinted in the endosperm of seeds were imprinted across cultivars, extending their functions to chromatin and transcriptional regulation, development, and signaling. Only 4% to 11% of the imprinted genes showed divergent imprinting.

Imprinted gene expression affects mostly single genes or groups of genes. Most of them are maternally expressed and inherited [36,40,41]. The best recognized is the maternal effect of genes during embryo development [42]. Early development in Arabidopsis is coordinated by the supply of auxin from the mother integuments of the ovule, which is required for the correct embryo development of embryos [40]. The genomic imprinting of the cereal endosperm influences the timing of endosperm cellularization [43]. An example of imprinted maternally expressed genes in cereals is a polycomb group, which is important for the cellularization of endosperm in rice [44]. Reciprocal crosses between tetraploid and hexaploid wheats showed that imprinted genes were identified in endosperm and embryo tissue, supporting the predominant maternal effect on early grain development [45]. Paternally expressed imprinted genes were associated with hybrid seed lethality in Capsella [46]. In maize, the *Dosage-effect defective1* (*ded1*) locus that contributes to seed size was found to be paternally imprinted [47]. The gene encodes a transcription factor that is specifically expressed during early endosperm development. There is also evidence that small RNAs might determine the paternal methylome by silencing transposons [48]. In addition to these reports, it is very difficult to find examples of paternally inherited genes, especially in cereals. Many studies have indicated dynamic changes in the epigenetic state, including DNA methylation, chromatin modifications, and small RNAs, which are observed during the reproductive development of plants [49,50]. The spatiotemporal pattern of gene expression, imprinting, and seed development in Arabidopsis endosperm is predominantly regulated by small maternal RNAs; however, they also originate from the paternal genome and the seed coat [51]. As reported by Tuteja et al. [52], imprinted paternally expressed genes, but not maternally expressed genes, in Arabidopsis evolve under positive Darwinian selection. These genes were involved in seed development processes, such as auxin biosynthesis and epigenetic regulation. Imprinted paternally expressed genes are mainly associated with hypomethylated maternal DNA alleles, which can be repressed by small genic RNAs and rarer with transposable elements [49,53]. Epigenetic changes can be developmentally regulated (developmental epigenetics). The state in which changes in DNA methylation are stable between generations and heritable is called transgenerational epigenetics [54].

Several *TaCKX* GFMs and *TaNAC2-5A* (*NAC2*) were previously selected as important regulators of yield-related traits. To determine how the expression patterns of selected genes are inherited in the developing spikes and seedling roots of the parents and the F_2_ generation, we used a reciprocal crossing strategy. The research hypothesis assumed that knowledge of inheritance of gene expression patterns that regulated yield-related traits indicated the way of selection of genotypes in wheat breeding. There is a research gap in documenting the inheritance of expression patterns for yield-related genes. We found that most of the genes in the F_2_ generation were expressed in a pater-of-origin-specific manner, which shed new light on the ways of selecting wheat lines and the breeding strategy.

## 2. Results

### 2.1. Reciprocal Crosses Indicate That the Expression Patterns of Most of the TaCKX GFM and Yield-Related Traits Are Mainly Inherited from the Male Parent

Relative values (related to the female parent = 1.0) of the expression profiles of *TaCKX* GFM and *NAC2* in 7 DAP spikes, seedling roots, and phenotypic traits in the female parent, male parent, and their six F_2_ progeny from one reciprocal cross of S12B × S6C (C1) and S6C × S12B (C2) are presented in Figure 1. The same data obtained in reciprocal crosses of D16 × KOH7 (C3) and KOH7 × D16 (C4); D19 × D16 (C5) and D16 × D19 (C6); D19 × KOH7 (C7) and KOH7 × D19 (C8) are visualized in Appendix A. S6C, the paternal parent of the S12B × S6C cross (C1), showed higher expression of *TaCKX1* and *NAC2* and lower expression of *TaCKX5* and *TaCKX10* in spikes than the maternal parent (S12B). The expression of *TaCKX1*, *NAC2*, *TaCKX5,* and *TaCKX11* in the spikes of the F_2_ progeny of this cross was higher (Figure 1A). In the reverse cross (S6C × S12B), when S12B was a paternal component (Figure 1B), *TaCKX1* and *NAC2* were expressed at low levels, *TaCKX5* and *10* were highly expressed in spikes compared to the maternal parent (S6C), and in F_2,_ *TaCKX1* and *NAC2* were expressed at low levels, and *TaCKX9*, *10*, *11*, and *5* were upregulated. In seedling roots, the paternal component of S12B × S6C (Figure 1A) showed high expression of *TaCKX5* and *NAC2* and low expression of *TaCKX10* and *11* in the parental parent compared to the maternal parent. In the roots of the F_2_ progeny of this cross, *TaCKX5* and *NAC2* were highly expressed, and *TaCKX11* was downregulated. The expression data in the parents of the reverse cross, in which S12B was the paternal component, were opposite. Their F_2_ progeny showed a strong upregulation of *TaCKX8*, *10*, and *11* and a downregulation of *TaCKX1*, *TaCKX3*, and *NAC2* in seedling roots. The total grain yield and the number of seeds in F_2_ of S12B × S6C were low, similar to the male parent (Figure 1A). The root weight in the F_2_ progeny of the same cross (S12B × S6C) was lower than that in the parents. The same yield components in the opposite cross (S6C × S12B) were high in one F_2_ sibling, comparable to the maternal parent in two progeny and lower than in the parents in three of them (Figure 1B). Interestingly, the root mass in F_2_ was higher than that in both parents.

As presented in Table 1 by colours, most of the expression patterns tested for *TaCKX* GFM and *NAC2* are inherited from the male parent (red). For example, up-regulated in 7 DAP spikes of the paternal parent *TaCKX1* and *NAC2* compared to the maternal parent is up-regulated in F_2_ as well. The upregulated *TaCKX5* and *NAC2* and downregulated *TaCKX11* in seedling roots of the paternal parents are similarly expressed in an F_2_. To summarize, the expression levels of all tested *TaCKX* GFMs and *NAC2* in 7 DAP spikes, in addition to being represented in different crosses, showed similar expression patterns to the paternal parents and were independent of the cross path. Among the *TaCKX* GFMs in 7 DAP spikes, which showed the paternal expression patterns were *TaCKX1*, *2.1*, *2.2.2*, *5*, *9*, *10*, and *11*. In the seedling roots, there were *TaCKX5*, *11*, *NAC2*; *TaCKX10*, *11*, *NAC2*; *TaCKX1*, *11*, *NAC2*; *TaCKX10*; *1*, *3*, *5*, *8*, *10*, *11*, *NAC2*; *TaCKX1*, *8*, *10*, *NAC2*; and *3*, *5*, *8*, *11*; *NAC2* (all tested but represented in different crosses). The only exceptions are *TaCKX5* in 7 DAP spikes of S12B × S6C (C1) and *TaCKX3* in seedling roots of KOH7 × D16 (C4), whose expression level is similar to that of the maternal parent (green).

Yield-related traits are represented by total grain yield and root mass (Table 1). Interestingly, grain yield in 7 out of 8 crosses is inherited from the paternal parent. The exceptions are the F_2_ progeny of S6C × S12B (C2), which show very large differences in yield, exceeding parental data. The root mass in F_2_ was lower than that in both parents or higher than that in both parents. In the first case, the root mass in the paternal parent was higher than that in the maternal parent, and in the second, the root mass in the paternal parent was lower than that in the maternal parent.

The results of crossing the low-yielding maternal parent with the higher-yielding paternal parent and their accompanying up- or down-regulated *TaCKX* GFMs and *NAC2* in F_2_ generations are presented in Figure 2.

Depending on the crosses, downregulation of *TaCKX5* with *TaCKX9* and upregulation of *NAC2* in spikes of the low-yielding maternal parent and the opposite regulation of these genes in spikes of the higher-yielding paternal parent resulted in high-yielding F_2_, characterized, as in the paternal component, by a higher expression level of *TaCKX9* and a lower expression level of *NAC2*. Upregulation of *TaCKX2.1* and *11* in spikes of the maternal parent and downregulation of these genes in the paternal parent were associated with downregulation of *TaCKX11* in high-yielding F_2_. Similarly, the upregulation of *TaCKX2.2.2* and the downregulation of *TaCKX10* in the spikes of the low-yielding maternal parent and opposite regulation of these genes, and the yield in the paternal parent, resulted in the downregulation of *TaCKX2.2.2* and the upregulation of *TaCKX10* in the spikes of the high-yielding F_2_. The expression of *TaCKX3* in seedling roots of the high-yielding paternal parent and F_2_ was upregulated. However, *TaCKX8* expression was upregulated, and *NAC2* was downregulated in the same organ of the paternal parent, but these genes were up- or down-regulated in F_2_, depending on the cross.

### 2.2. Cooperating and Opposite-Functioning Genes

*TaCKX5* with *TaCKX9* (yellow) and *TaCKX2.1* with *TaCKX11* (green) showed coordinated up- or downregulation in 7 DAP spikes of the paternal parent of C1, C2, C6, and C8 crosses; and C3, C4, C7, and C8 crosses, respectively (Table 2). Higher expression of *TaCKX5* and *9* in this parent was associated with a higher yield in F_2_. However, a higher coordinated expression of *TaCKX2.1* with *TaCKX11* in the paternal parent determined a lower yield in F_2_, and, in contrast, a lower expression of these two genes in the paternal parent was associated with a higher yield.

In the 7 DAP spikes, the paternal parent of the C3, C4, C5, and C6 crosses, *TaCKX2.2.2,* showed opposite expression to *TaCKX10* (blue), and upregulation of the first and downregulation of the second were associated with lower yield (but not in C6). In the paternal parent of the C1, C2, C7, and C8 crosses, *NAC2* was oppositely expressed to *TaCKX9*; in these crosses, a high yield was observed when *TaCKX9* was upregulated and *NAC2* was downregulated, and vice versa (only in C7 and C8). Furthermore, upregulated *TaCKX5* and downregulated *TaCKX1* were associated with high root mass in C2 and conversely in the reverse cross (C1).

Among the *TaCKX* genes coordinately expressed in the paternal seedling roots were *TaCKX3*, *5,* and 8 (green) in C5, C6, C7, and C8 crosses; *TaCKX3* and *8* (green) in C1 and C2 crosses; *TaCKX10*, *11*, and *1* (yellow), and *NAC2* in C3 to C7 crosses; and *TaCKX10* and *11* (yellow) in C1 and C2 crosses. However, in one reciprocal cross, C3 and C4, the expression of *TaCKX3* and *TaCKX5* was opposite, and in the case of upregulation of *TaCKX3* and downregulation of *TaCKX5*, the grain yield in F_2_ was higher.

In three reciprocal crosses, *NAC2* was downregulated in paternal roots (C2, C4, and C8), and in two of them (C2 and C8), *NAC2* was downregulated in paternal spikes as well. This negative regulation of *NAC2* occurred in the F_2_ progeny, which was accompanied by a higher yield and a higher or similar to the parents’ mass of the seedling roots. In contrast, in another way crosses, when expression of *NAC2* was increased in paternal roots (C1, C3, and C7) and was upregulated in paternal spikes, the same was observed in F_2_ progeny, characterized by lower yield and lower or similar to the parent mass of the roots.

The higher yield in F_2_ has been associated with the same or higher CKX activity, as in the paternal parent, in 7 DAP spikes. A higher number of semi-empty spikes, which occurred in low-yielding F_2_ of the C3, C5, and C7 crosses, was accompanied by downregulated *TaCKX10* and/or upregulated *TaCKX11* in 7 DAP spikes and upregulated *TaCKX10*, *11,* and *NAC2* or downregulated *TaCKX10* and *NAC2* in seedling roots.

### 2.3. The Correlation Coefficients between TaCKX GFMs and NAC2 Expression, CKX Activity, and Yield-Related Traits Were Significant for Both Parents or the Maternal or Paternal Parent Separately

The correlation coefficients between *TaCKX* GFMs and *NAC2* expression, CKX activity, and yield-related traits in reciprocal crosses were analyzed separately for the maternal parent and F_2_, paternal parent, and F_2_ for each cross (Appendix A).

### 2.4. Correlations between TaCKX GFM and NAC2 Expression and Yield-Related Traits in the Group of Maternal Plants, and F_2_ and Paternal Plants, and F_2_ of Reciprocal Crosses

Seed number and spike number were positively correlated (Table 3 and Appendix A); however, each of these yield-related traits was correlated with different *TaCKX* GFMs.

#### 2.4.1. Seed Number

The decrease in seed number in maternal plants and their F_2_ (M and F_2_) of the C1 cross (S12B × S6C) was strongly negatively correlated with upregulated *TaCKX1* and *TaCKX5* in spikes and positively correlated with the downregulated *TaCKX11* in the seedling roots of F_2_. There was no significant correlation between the expression of *TaCKX* GFM and the yield-related traits in the groups of paternal plants (P) and F_2_ in the same cross, and M and F_2_, and P and F_2_ in the reverse, C2 cross. The F_2_ progeny in this reverse cross showed a similar yield and greater root mass compared to the parents.

In both M and F_2_, and P and F_2_ of C3, the decrease in seed number was strongly positively correlated with *TaCKX2.1* and *TaCKX2.2.2* in spikes and positively correlated with *TaCKX3* and *TaCKX8* but negatively correlated with *TaCKX11* in seedling roots. These correlations were not significant in the reverse, C4 cross, in which the M (KOH7) and F_2_ plants showed higher yields.

The decrease in seed number in M × F_2_ of C5 was negatively correlated with *TaCKX1* in spikes and positively correlated with downregulated *TaCKX3* and *5,* and upregulated *NAC2* in seedling roots. There were also positive correlations of *TaCKX3* in roots between P and F_2_ of the same cross. These correlations were not significant in the reverse C6 cross; however, F_2_ of this cross was characterized by higher yield and similar root mass than in the parents.

The increase in seed number was strongly positively correlated with downregulated *TaCKX2.1* in spikes and negatively correlated with downregulated *TaCKX1* in the seedling roots only in F_2_ progeny of a C8 cross, 14K (in the case of P × F_2_ only for *TaCKX2.1*).

#### 2.4.2. Spike Number

The decrease in the number of spikes in M × F_2_ and P × F_2_ of the C1 cross was strongly positively correlated with *TaCKX2.1* and *TaCKX2.2.2* in spikes and positively correlated with *TaCKX11* in seedling roots (only in M × F_2_). There was also a significant and positive correlation between the expression of *TaCKX2*.*2.2* and the number of spikes in the reverse C2 cross of M and F_2_. Furthermore, in the same cross, the spike number was negatively correlated with downregulated *NAC2*, but only in the P × F_2_ group. There were no significant correlations between *TaCKX* GFM expression and spike number in M × F_2_ and P × F_2_ spikes of C3 and C4. However, there was a strong and positive correlation of the spike number with *TaCKX8* in the roots of C3 and *TaCKX5* in the roots of C4.

The decreased spike number in M × F_2_ and P × F_2_ of C5 was not correlated with any *TaCKX* expressed in the spikes but was negatively correlated with *TaCKX1*, positively correlated with *TaCKX5,* and positively correlated with *NAC2* expressed in the seedling roots. Conversely, in reverse C6 cross, there was a positive correlation of the spike number with *TaCKX2.1* in a P × F_2_, which resulted in a higher yield phenotype in the F_2_.

The spike number in C7 and C8 crosses was not correlated with the level of expression of any gene tested in the spikes; however, it was negatively correlated with the expression of *TaCKX1*, 5, and *8* in seedling roots.

#### 2.4.3. TGW

TGW was positively correlated with *TaCKX2.1*, *10,* and *NAC2* in spikes of M and F_2_ of C1 and with *TaCKX2.2.2* of P and F_2_ of the same cross. There was no correlation in F_2_ between the expression of the genes tested and TGW in spikes of the C2 and roots of the C1 and C2 crosses. There were no correlations between the TGW and *TaCKX* genes in the spikes and roots of C3. However, there was a negative correlation of this trait with *TaCKX2.2.2* in spikes and positive correlations with *TaCKX10* and *NAC2* in roots of the reciprocal C4 cross. Positive correlations of *NAC2* with TGW were also observed in the roots of C5 but not in those of C6. Negative correlations of *TaCKX2.1* and *2.2.2* in spikes with TGW were observed in both reciprocal crosses, C7 and C8. Additionally, *TaCKX9* was negatively correlated with the trait in M + F_2,_ *TaCKX5* was positively correlated with the trait in P + F_2_ of C7, and *TaCKX10* was negatively correlated with TGW in P + F_2_ of C8. There was no correlation between TGW and any gene expression in roots.

#### 2.4.4. Root Mass

The mass is positively correlated with the expression of *TaCKX5*, *11,* and *NAC2* in spikes of M + F_2_ of C1 and negatively correlated with *NAC2* in M + F_2_ of C2. A positive correlation between root mass and *TaCKX11,* and *NAC2* was also visible in P + F_2_ of C3, and a negative correlation between trait and *NAC2* expression was also observed in spikes of M + F_2_ of C5 and P + F_2_ of C7. Furthermore, in the C1 cross, this trait was negatively correlated with the expression of *TaCKX1*, *3*, *10,* and *NAC2* in roots of M + F_2_ and with *TaCKX11* in roots of P + F_2_. There were also positive correlations between root mass and *TaCKX1* (M + P of C3), root mass and *TaCKX2.1* (P + F_2_ of C4; P + F_2_ of C_5_; M + F_2_ of C6), and root mass and *TaCKX2.2.2* (P + F_2_ of C4; M + F_2_ and P + F_2_ of C6). The expression of another gene, *TaCKX10,* in spikes, was positively correlated with root mass in P + F_2_ of C4 but negatively correlated in M + F_2_ of C8. Correlations between root mass and gene expression tested in roots were dependent on the parent and cross. There were negative correlations with *TaCKX1* in 5 out of 16 combinations tested, negative correlations with *TaCKX3* in 3 combinations, but positive correlations in two combinations, positive correlations with *TaCKX5* in two combinations, and single positive or negative correlations with *TaCKX8*, *11,* and *NAC2*. The root mass in single combinations was positively correlated with the yield (twice), height of the plant (once), and length of the spike (once), and negatively correlated with seed number of seeds (twice).

#### 2.4.5. Semi-Empty Spikes

The number of semi-empty spikes was positively correlated with the expression of *TaCKX9* in the P + F_2_ C1, C2, and M + F_2_ C3 crosses, positively correlated with *TaCKX5* in the M + F_2_, and P + F_2_ C1 and C6 crosses, and positively correlated with *TaCKX10* in the M + F_2_ C7 and P + F_2_ C8 crosses, all expressed in 7 DAP spikes. The negative correlation between the number of semi-empty spikes and *TaCKX2.1* was in P + F_2_ of C5, and between the same trait and *TaCKX11* was in P + F_2_ of C3. In seedling roots of various crosses, this trait was mainly negatively correlated with *TaCKX5*, *8*, *10,* and *NAC2*.

Generally, negative correlations between the expression of *TaCKX2.1*, *2.2.2,* and *10* in spikes and TGW, seed number, seed yield, and spike number were correlated with higher yield, and positive correlations were correlated with lower yield. On the other hand, positive correlations between the expression of these genes and root mass determine a higher yield in F_2_. Higher yield in F_2_ is also associated with balanced CKX enzyme activity in spikes and seedling roots.

A summary of the regulation of yield-related traits by *TaCKX* GFMs and *NAC2* in the high-yielding progeny of F_2_ is presented in Figure 3.

## 3. Discussion

Common wheat is a very important cereal crop for feeding the world’s population; therefore, continued improvement of the yield of this species is significant. *CKX* GFMs have already been documented to perform a pivotal role in determining yield-related traits in many plant species, including wheat [4,12]. The genes are tissue-specific; they encode cytokinin oxidase/dehydrogenase, the enzyme that irreversibly degrades cytokinins. We have already characterized the role of *TaCKX1* and *TaCKX2* in the regulation of yield traits in awnless and owned-spike cultivars [5,6,7]. The range of natural variation in the expression levels of most *TaCKX* genes among breeding lines and cultivars was very high, indicating the possibility of selecting beneficial genotypes for breeding purposes [8]. Therefore, we were interested in how the expression of these genes is inherited.

### 3.1. The Expression Patterns of Most TaCKX GFMs and TaNAC2-5A Are Mainly Inherited from the Paternal Parent

Comparison of the expression patterns of most of the *TaCKX* GFMs and yield-related traits between parents and F_2_ progeny in all reciprocal crosses tested indicated their inheritance from the paternal parent. This rule includes expression patterns in both tissues tested, 7 DAP spikes, and seedling roots, and all *TaCKX* GFMs and *TaNAC2-5A* tested were represented in different crosses. The exception was *TaCKX5* expressed in 7 DAP spikes, and *TaCKX3* expressed in seedling roots, for which the expression level in single crosses was inherited from the maternal parent. Furthermore, high or low yield was predominantly inherited from the paternal parent, and root mass was inherited from both parents or in one reciprocal cross from the maternal parent. We have not found such examples of inheritance in the literature; however, some deviations from parental additivity of expression in polyploid plants were described [2]. An example of such non-additive gene expression takes place when the gene expression level in progeny is higher than that of one parent. The expression level dominance of one parent, also called genomic imprinting, is epigenetic in origin and was investigated primarily at the molecular level in plants and animals [36,37]. The main regulators of gene imprinting are DNA and histone methylation asymmetries between parental genomes. Most of the imprinted genes in the endosperm of grains of different rice cultivars are imprinted across cultivars, and their functions are associated with the regulation of transcription, development, and signaling [39]. Imprinting might affect a single gene or a group of genes. Genes that showed conserved imprinting in cereals have been shown to reveal positive selection and were suggested to regulate seed development in a dose-dependent manner [38]. The only example of a paternally imprinted locus in maize is *ded1*, which encodes a transcription factor specifically expressed during early embryo development and activates early embryo genes that contribute to grain set and weight [47]. To our knowledge, there are no examples of paternally inherited expression patterns. According to Arabidopsis research, imprinted paternally expressed genes during seed development are mainly related to hypomethylated maternal alleles, repressed by small RNAs or less frequently with transposable elements [48,49]. Contrary to developmental epigenetics, in the case of transgenerational epigenetics, these epigenetic changes do not reset between generations, and this type of inheritance is more related to plants than animals (heritable changes in DNA methylation) [54]. Therefore, we suggest that this paternal inheritance of selected *TaCKX* GFMs is an effect of transgenerational epigenetic changes, not reset between generations. These heritable epigenetic changes might be effects of DNA methylation, repression of maternal alleles by small RNAs, transposable elements, or, most likely, transcription factors. From our in silico analysis and expression analysis (Iqbal et al., not published yet), several NAC transcription factors appear to strongly regulate the expression of *TaCKX* GFMs and *TaIPT* GFMs, influencing yield-related traits [24].

### 3.2. Cooperation of TaCKX GFMs and TaNAC2-5A in the Determination of Yield-Related Traits

The coordinated high or low level of expression of a few groups of genes in the paternal parent positively or negatively regulates higher or lower yield. In two reciprocal crosses, where both *TaCKX5* and *TACKX9* showed high expression in 7 DAP spikes of the paternal parent, the yield in the F_2_ progeny was high and vice versa. In others, the high yield in the F_2_ progeny was determined by a low level of expression of *TaCKX2.1* and *TaCKX11* in spikes of the paternal parent and high levels of their expression in the maternal parent. The level of expression of *TaNAC2-5A* in the paternal parent and/or F_2_ was in opposition to *TaCKX5* and *9*; however, it was in agreement with *TaCKX11* and *TaCKX2.1,* suggesting their role in the regulation of transcription of these genes. In fact, it was proven by correlation analysis of its expression with yield-related traits [8]. Opposite cooperation of some of the genes in paternal spikes, which resulted in high or low yield in F_2_, has also been observed. The high level of expression of *TaCKX2.2.2* and the low level of expression of *TaCKX10* predominantly resulted in the low yield in F_2_ progeny and vice versa.

Such common rules of gene expression in the paternal parent associated with yield in the paternal parent and F_2_ progeny were also observed in the seedling roots. The high level of expression of *TaCKX3* and *TaCKX8* in the paternal parents of three reciprocal crosses resulted in high yield in the F_2_ progeny and vice versa. The expression of *TaNAC2-5A* in spikes and seedling roots of high-yielding paternal parents and F_2_ progeny showed a predominantly low expression level and inversely. These principles of paternal inheritance of selected *TaCKX* GFMs and *TaNAC2-5A* expression associated with high yield could be directly involved as molecular markers in high-yielding wheat breeding.

### 3.3. Regulation of Yield-Related Traits by TaCKX GFMs and TaNAC2-5A in the F_2_ Generation

The grain number, grain yield, spike number, and TGW were strongly positively correlated with *TaCKX2.1* and *TaCKX2.2.2* independent of the parent; however, only in the crosses resulted in decreased yield. In contrast, negative correlations were observed between *TaCKX2.1*, *TaCKX2.2.2,* and TGW in a reciprocal cross of C7/C8 and *TaCKX2.2.2* in a one-way cross (C4), in which the F_2_ progeny had a higher yield. All these correlations prove our earlier observations [6,7]. Modified wheat lines with 60% decreased expression of *TaCKX2.2.2,* and a slight decrease in the *TaCKX2.2.1* and *2.1* genes exhibit a significantly higher TGW and slightly increased yield [7]. Interestingly, this result was observed in cultivars and breeding lines that represent awnless spikes. In the owned-spike cultivar, silencing of the *TaCKX2* genes co-expressed with other *TaCKX* resulted in decreased yield; however, TGW was at the same level as in non-silent plants [5]. Furthermore, a strong feedback mechanism for regulation of the expression of *TaCKX2* and *TaCKX1* genes was observed in both awnless and owned-spike cultivars [5,6,7]. Silencing of *TaCKX2* genes upregulated the expression of *TaCKX1* and vice versa. This feedback mechanism could explain the observed positive correlations of the *TaCKX2* genes with yield-related traits in low-yielding F_2_ progeny and negative correlations in high-yielding F_2_ progeny. A similar mechanism is visible when we analyze individual traits in high-yielding F_2_, such as grain number, grain yield, and spike number. These traits are promoted by up-regulated in 7 DAP spikes *TaCKX2.1* and down-regulated *TaCKX1*. Silencing of *HvCKX1* in barley, which is an ortholog of *TaCKX1*, decreased CKX enzyme activity and led to increased seedling root mass and higher plant productivity [55]; however, knock-out of this gene caused a significant decrease in CKX enzyme activity but no changes in grain yield were observed [56]. These differences might be explained by differences in the level of decreased gene expression, which variously coordinate the expression of other genes, regulate phytohormone levels, and determine particular phenotypes, as was already documented in wheat [5,7].

The association of *TaCKX2* genes with yield-related traits has also been reported in different wheat cultivars or genotypes. Zhang et al. [20] showed that *TaCKX6* (renamed by Chen et al. [11] *TaCKX2.2.1-3D*), which is an ortholog of rice *OsCKX2* associated with grain number [57], is related to grain weight. Another allele of *CKX2*, *TaCKX6a02* [21], annotated as *TACKX2.1* [58], significantly correlated with grain size, weight, and grain filling rate. Wheat plants with silenced by RNAi expression of *TaCKX2.2.1-3A* (originally *TaCKX2.4*) showed a strong correlation with the number of grains per spike implied by more filled florets [59]. Since *TaCKX2.2.1-3D* was associated with grain weight, these differences in functions between *TaCKX2.2.1-3A* and *TaCKX2.2.1-3D* were interpreted as subgenome-dependent.

Grain number was also negatively correlated with *TaCKX1* and *TaCKX5*, and grain yield was negatively correlated with *TaCKX1*, predominantly for M and F_2_, which were characterized by decreased grain number and lower yield. This observation is also in agreement with previous research. The silencing of *TaCKX1* caused an increase in spike number and grain number but a decrease in TGW because this trait is opposite to grain number [6]. The low-yield progeny of F_2_ showed positive correlations between the expression of *TaCKX11*, *3*, *5*, and *8*, and *NAC2* in the seedling roots and the grain number, the spike number and the grain yield; however, the higher-yield progeny of F_2_ displayed a negative correlation between *TaCKX1* and these yield-related traits in the seedling roots of some crosses. In summary, high-yielding F_2_ was the result of upregulation of *TaCKX2.1* in spikes and downregulation of *TaCKX1* in seedling roots. As documented earlier, *TaCKX11*, *5*, *8* and *TaNAC2-5A* are expressed in all organs, and their expression is correlated with the expression of spike-specific *TaCKX2* and *TaCKX1* [8,58].

Rice *OsCKX11* is an orthologue of wheat *TaCKX11* and is highly expressed in the roots, leaves, and panicles. The gene was shown to coordinate the simultaneous regulation of leaf senescence and grain number by the relationship of source and sink [60]. Since *TaCKX11* is expressed in seedling roots and highly expressed in leaves, inflorescences, and 0, 7, and 14 DAP spikes, it could perform a similar function. This is partly proven by silencing of the *TaCKX2* genes in awnless spikes of cv. Kontesa, which resulted in significant upregulation of *TaCKX11* and growth of TGW, and chlorophyll content in flag leaves [7]. In contrast, *TaCKX11* is significantly negatively regulated by *TaCKX1*, resulting in a higher spike number and grain number [6]. Its orthologue in rice, *OsCKX11,* was found to regulate leaf senescence and grain number by the coordinated source and sink relationship [60].

Based on a summary of the regulation of yield-related traits in high-yielding F_2_, it is possible to identify singular genes or groups of genes that are up- or down-regulated in 7 DAP spike or seedling roots and specifically regulate yield-related traits. Upregulated in spikes *TaCKX2.1* and downregulated in seedling roots *TaCKX1* were found to determine grain number, grain yield, spike number, and spike length. Furthermore, upregulated in 7 DAP spikes *TaCKX10* and downregulated *TaNAC2-5A,* together with others, depending on cross, control spike length, semi-empty spikes, root mass, and increased grain yield. As discussed above, high TGW is in contrary to high grain number and partly grain yield and was strongly determined by downregulated *TaCKX2.2.2* together with *TaCKX2.1* in 7 DAP spikes and upregulated *TaCKX10* and *NAC2* in seedling roots. The upregulated in seedling roots *TaNAC2-5A* participates in the determination of TGW and plant height, and the downregulation of *TaNAC2-5A* in seedling roots controls the development of semi-empty spikes and root mass.

In previous research, *TaNAC2-5A* has been documented as a gene encoding a nitrate-inducible wheat transcription factor. Overexpression of the gene improved root growth, grain yield, and grain nitrate concentration [25]. This is in agreement with our observations of growth of TGW but not enhanced roots. The increase was argued to be the consequence of regulation of nitrate concentration and its remobilization in developing grains by direct binding of the *TaNAC2-5A* protein to the promoter of the nitrate transporter, *TaNRT2.5-3A* and positive regulation of its expression [61]. The expression of *TaNAC2-5A* is coregulated by expressed in 7 DAP spikes *TaCKX2* genes and expressed in 7 DAP spikes and seedling roots *TaCKX1* gene [5,7]. Independent of awnless or awned-spike genotype, downregulation of *TaCKX2* genes by RNAi significantly increased *TaNAC2-5A* expression, resulting in higher chlorophyll content in flag leaves and delayed leaf senescence. As discussed above, the strong feedback mechanism between the *TaCKX2* and *TaCKX1* genes implies that downregulation of *TaCKX1* resulted in opposite results. Similar to our observations in wheat, an ortholog of *TaNAC2-5A* in rice, *OsNAC2,* was described as a negative regulator of crown root number and root length [62]. Its expression was positively correlated with cytokinin synthesis genes, *OsIPT3*, *5*, the gene determining the formation of active cytokinins, *OsLOG3*, and negatively correlated with *OsCKX4* and *5*. The authors concluded that *OsNAC2* stimulated cytokinin accumulation by suppressing *CKX* expression and stimulating *IPT* expression by binding the OsNAC protein to the promoters of these genes. Therefore, *OsNAC2* functions as an integrator of cytokinin and auxin signals that regulate root growth. In our experiments, orthologous to *OsCKX4*, *TaCKX4* was not tested due to its weak expression in roots. However, the up-regulated expression of highly specific in seedling roots *TaCKX3* and *TaCKX8* [8,58] was antagonistically regulated by *TaNAC2-5A* in these organs, positively influencing seedling growth. Furthermore, our in silico analysis of *TaNACs* with *TaIPTs* and *TaCKXs* showed that the same NAC proteins might join promotor sites of cytokinin synthesis and cytokinin degradation genes in wheat (Iqbal et al., not published yet).

## 4. Materials and Methods

### 4.1. Plant Material

Five common wheat breeding lines and cultivars (*Triticum aestivum* L.), named S12B, S6C, D16, KOH7, and D19, which showed differences in the expression levels of *TaCKX* GFMs and *TaNAC2-5A* (*NAC2*) in 7 DAP spikes, seedling roots, and yield-related traits were selected for the study. They were used in four reciprocal crosses: (1) S12B × S6C and S6C × S12B (C1 and C2, respectively); (2) D16 × KOH7 and KOH7 × D16 (C3 and C4, respectively); (3) D19 × D16 and D16 × D19 (C5 and C6, respectively); and (4) D19 × KOH7 and KOH7 × D19 (C7 and C8, respectively) to obtain the F_1_ and F_2_ generations. The experimental tissue samples were collected from the parental lines and their F_2_ progeny growing in a growth chamber during the same period.

Ten seeds of each genotype germinated in Petri dishes for five days at room temperature in the dark. Six out of ten seedlings from each Petri dish were replanted in pots with soil. The plants were grown in a growth chamber under controlled environmental conditions with 20 °C day/18 °C night temperatures and a 16 h light/8 h dark photoperiod. The light intensity was 350 μmol s^−1^·m^−2^. The plants were irrigated three times a week and fertilized once a week with Florovit according to the manufacturer’s instructions.

The following tissue samples in three biological replicates were collected: 5-day-old seedling roots, which were cut 0.5 cm from the root base before replanting in the pots, and first 7 DAP) spikes from the same plants grown in the growth chamber. All of these samples were collected at 9:00 a.m. The collected material was frozen in liquid nitrogen and kept at −80 °C until use.

### 4.2. Cross-Breeding

The maternal plant was deprived of its own anthers so that it would not self-fertilize, then pollinated by transferring three anthers from the paternal plant for each ovary of the maternal parent plant and placed in an isolator. The seeds were harvested.

### 4.3. RNA Extraction and cDNA Synthesis

Total RNA from 7 DAP spikes and roots from 5-day-old seedlings was extracted using TRI Reagent (Invitrogen, Lithuana) according to the manufacturer’s protocol. The concentration and purity of the isolated RNA were determined using a NanoDrop spectrophotometer (NanoDrop ND-1000, Thermi Fisher Scientific, Wilmington, DE, USA), and the integrity was checked on 1.5% (*w/v*) agarose gels. To remove residual DNA, RNA samples were treated with DNase I (Thermo Fisher Scientific, Lithuana). Each time, 1 μg of good quality RNA was used for cDNA synthesis using the RevertAid First Strand cDNA Synthesis Kit (Thermo Fisher Scientific, Lithuana) following the manufacturer’s instructions. The cDNA was diluted 20 times prior to use in the RT-qPCR assays.

### 4.4. Quantitative RT-qPCR

RT-qPCR assays were performed for 10 genes: *TaCKX1*, *TaCKX2.1*, *TaCKX2.2.2*, *TaCKX3*, *TaCKX5*, *TaCKX8*, *TaCKX9*, *TaCKX10*, *TaCKX11*, and *TaNAC2-5A*. The sequences of the primers for each gene are shown in Appendix A. All real-time reactions were performed on a Rotor-Gene Q (QIAGEN Hilden, Germany) thermal cycler using 1× HOT FIREPol EvaGreen qPCR Mix Plus (Solis BioDyne, Estonia), 0.2 μM of each primer and 4 μL of cDNA in a total volume of 10 μL. Each reaction was carried out in three biological and three technical replicates in the following temperature profile: initial denaturation and polymerase activation of 95 °C–12 min (95 °C–25 s, 62 °C–25 s, 72 °C–25 s) × 45 cycles, 72 °C–5 min, with melting curve at 72–99 °C 5 s per step. The expression of *TaCKX* genes was calculated according to the two standard curve method using *ADP-ribosylation factor* (*Ref 2*) as a normalizer. The relative expression for each *TaCKX* GFM and *TaNAC2-5A* was calculated in relation to the control female parents, set as 1.00.

### 4.5. Analysis of CKX Activity

CKX enzyme activity was performed in the same samples subjected to *TaCKX* gene expression analysis according to the procedure developed by Frebort et al. [63] and optimized for wheat tissues. The plant material was powdered with liquid nitrogen using a hand mortar and extracted with a 3-fold excess (*v*/*w*) of 0.2 M Tris–HCl buffer, pH 8.0, containing 1 mM phenylmethylsulfonyl fluoride (PMSF) and 0.3% Triton X-100 ((St. Louis, MO, USA). Plant samples were incubated in a reaction mixture consisting of 100 mM McIlvaine buffer, 0.25 mM of the electron acceptor dichlorophenolindophenol and 0.1 mM of substrate (N6-isopentenyl adenine). The volume of the enzyme sample used for the assay was adjusted based on the enzyme activity. The incubation temperature was 37 °C for 1–16 h. After incubation, the reaction was stopped by adding 0.3 mL of 40% trichloroacetic acid (TCA) and 0.2 mL of 2% 4-aminophenol (PAF). The product concentration was determined by scanning the absorption spectrum from 230 nm to 550 nm. The total protein concentration was estimated based on the standard curve of bovine serum albumin (BSA) according to the Bradford procedure [64].

### 4.6. Measurement of Yield-Related Traits

Morphometric measurement of yield-related traits of selected genotypes was performed. The described traits were plant height, spike number, semi-empty spike number, tiller number, spike length, grain yield, grain number, TGW, and 5-day seedling root weight.

### 4.7. Statistical Analysis

Statistical analysis was performed using Statistica 13 software (StatSoft). The normality of the data distribution was tested using the Shapiro–Wilk test. The significance of the changes was analyzed using ANOVA variance analysis and post hoc tests. The correlation coefficients were determined using parametric correlation matrices (Pearson’s test) or a nonparametric correlation (Spearman’s test).

## 5. Conclusions

We indicate, for the first time, that the pattern of expression of selected *TaCKX* GFMs and *TaNAC2-5A*, and grain yield in wheat, is paternally inherited by the F_2_ generation. Pater-origin transmission of gene expression levels sheds new light on the method of parent selection and crossing to obtain high-yielding phenotypes. We also showed which genes cooperate together by upregulation or downregulation and which function in the opposite manner in establishing yield-related traits. This knowledge can be applied to select the desirable phenotype in F_2_. For example, a high-yielding paternal parent with downregulated, compared to the maternal parent, expression of *TaCKX2.1* and *TaCKX11* in 7 DAP spikes and upregulated expression of *TaCKX3* and *TaCKX8* and downregulated *TaNAC2-5A* in seedling roots is expected to transmit this pattern of expression to F_2_, which will result in a high yield. The main problem is the antagonistic expression patterns of genes for some important yield-related traits, such as grain number, grain yield, and spike number, to TGW, which is the result of the feedback mechanism of the regulation of expression between *TaCKX1* and *TaCKX2* genes and others. The expression analysis of *TaNAC2-5A* and the in silico analysis of *TaNAC* GFMs revealed that the encoded proteins participate in the regulation of transcription of selected *TaCKX* genes responsible for cytokinin degradation and *TaIPT* genes responsible for cytokinin biosynthesis. Therefore, *TaNACs* are important additional regulators of yield-related traits in wheat, which should be taken into consideration in wheat breeding.

## 6. Patents

Nadolska-Orczyk A., Szala K., Dmochowska-Boguta M., Orczyk W. Wzory ekspresji genów jako nowe markery molekularne produktywności zbóż oraz sposób przekazywania wysokiej produktywności I strategia selekcji wysokoplonujących odmian zbóż. (Patterns of gene expression as new molecular markers of cereal productivity and a way of transfer of high yield and the strategy for selecting high-yielding cereal varieties). Patent application filed with the Polish Patent Office (UP RP) 23 January 2023, nr P.443557.

## Figures and Tables

**Figure 1 ijms-24-08196-f001:**
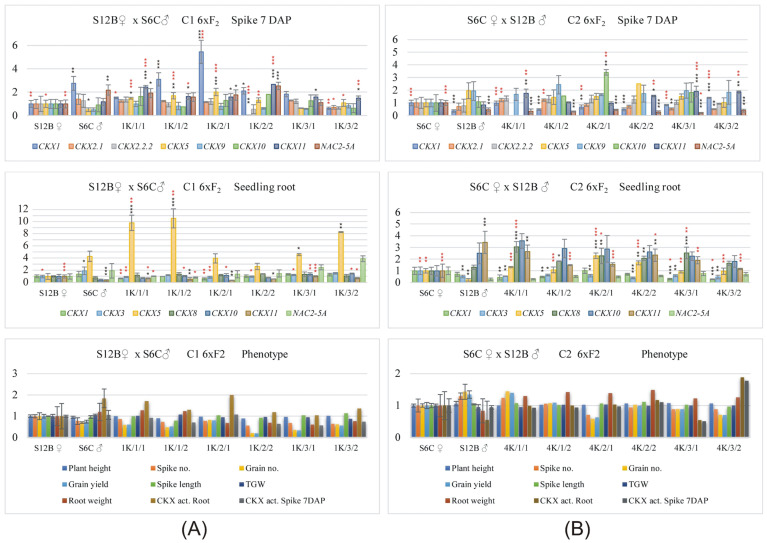
Example of *TaCKX* GFM and *NAC2* expression profiles in 7 DAP spikes, seedling roots, and phenotypic traits in the maternal parent, paternal parent, and their six F_2_ progeny, from reciprocal cresses of S12B × S6C, C1 (**A**) and S6C × S12B, C2 (**B**). The data represent mean values with standard deviation. Black and red asterisks indicate statistical significance compared to the maternal parent or paternal parent, respectively (* 0.05 > *p* ≥ 0.01, ** 0.01 > *p* ≥ 0.001, *** *p* < 0.001) using the ANOVA test followed by the LSD post hoc test (STATISTICA 10, StatSoft).

**Figure 2 ijms-24-08196-f002:**
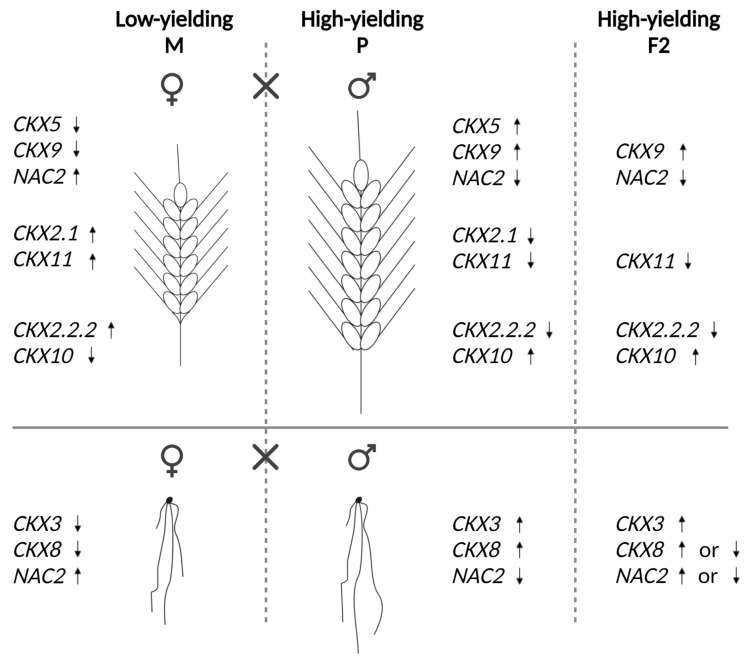
Models of up- (↑) or down-regulation (↓) of *TaCKX* GFMs and *NAC2* in low-yielding maternal parent crossed with higher-yielding paternal parent and their F_2_ progeny.

**Figure 3 ijms-24-08196-f003:**
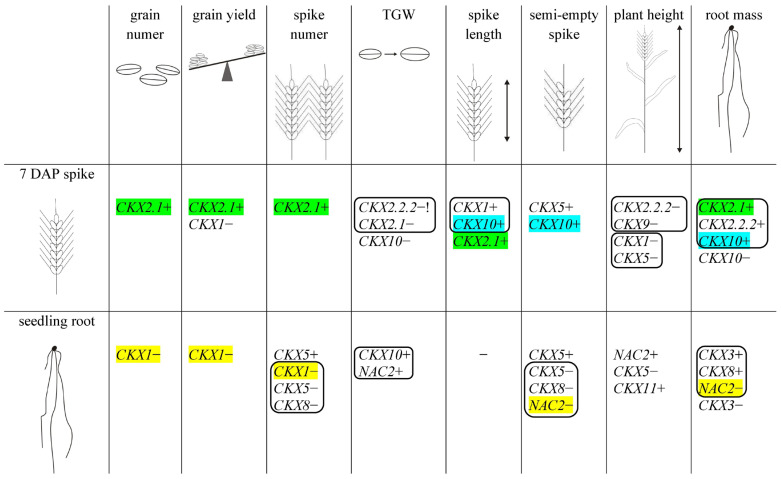
Regulation of yield-related traits by *TaCKX* GFMs and *NAC2* in the high-yielding progeny of F_2_ based on correlation coefficients.

**Table 1 ijms-24-08196-t001:** *TaCKX* GFMs and *NAC2* with high (↑), very high (↑↑), low (↓) or very low (↓↓) expression levels in 7 DAP spikes and seedling roots of the maternal parent (M), the paternal parent (P) and their F_2_ progeny from cresses of D16 × KOH7 (C3) and KOH7 × D16 (C4); D19 × D16 (C5) and D16 × D19 (C6); D19 × KOH7 (C7) and KOH7 × D19 (C8), and high (↑) and low (↓) parameters of yield and root mass. Character colours indicate similar patterns of gene expression and yield-related traits in F_2_ and paternal parent (red) or in F_2_ and maternal parent (green).

	M	P	F_2_
	**C1 = S12B × S6C**		
CKX expression 7 DAP	*CKX1*↓	* CKX1 * ↑	*CKX1*, *11*↑↑
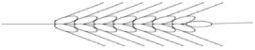	*NAC2*↓	* NAC2 * ↑	*CKX5*, *NAC2*↑
	*CKX5*, *9*↑	*CKX5*, *9*↓	
CKX expression root	*CKX5*, *NAC2*↓↓	* CKX5 * ↑↑	* CKX5 * ↑↑↑, *NAC2*↑
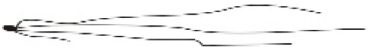	*CKX3*↓	*CKX3*↑, *NAC2*↑	
	*CKX11*, *10*↑	*CKX11*, *10*↓	* CKX11 * ↓
yield-related traits	yield↑	yield↓	yield↓↓
	CKX act. spike=	CKX act. spike=	CKX act. spike↓↓
	root=↓	root=↑	root=↓
	CKX act. root↓↓	CKX act. root↑↑	CKX act. root↑↑
	**C2 = S6C × S12B**		
CKX expression 7 DAP	*CKX5*, *9*↓	* CKX5 * , *9*↑	*CKX9*, *10*, *11*, *5*↑
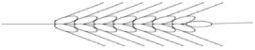	*CKX1*, *2.1*, *NAC2*↑	*CKX1*, *2.1*, *NAC2*↓	* CKX1 * , *NAC2*↓
CKX expression root	*CKX10*, *11*↓↓	* CKX10 * , *11*↑↑	*CKX5, 8*, *10*, *11*↑↑
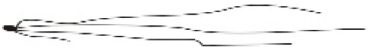	*CKX1, 3*, *5*, *NAC2*↑	*CKX1, 3*, *5*, *NAC2*↓	*CKX1*, *3*, *NAC2*↓
yield-related traits	yield↓	yield↑	yield=↓
	CKX act. spike=	CKX act. spike=	CKX act. Spike=
	root↑	root=↓	root↑↑
	CKX act. root↑	CKX act. root↓	CKX act. root=
	**C3 = D16 × KOH7**		
CKX expression 7 DAP	*CKX11*↓↓	*CKX11*↑↑	*CKX5, 9*↑↑
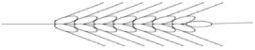	*CKX2.1*, *2.2.2*↓	*CKX2.1*, *2.2.2*↑	
	*CKX10*↑	*CKX10*↓	
CKX expression root	*NAC2*↓↓	* NAC2 * ↑↑	*CKX3*↓
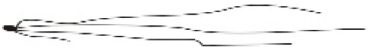	*CKX1*, *5*, *10*, *11*↓↓	*CKX1*, *5*, *10*, *11*↑↑	*CKX1*, *8*, *11*, *NAC2*↑↑
yield-related traits	yield↑	yield↓	yield↓
	CKX act. spike=↑	CKX act. spike=↓	CKX act. spike↓↓
	root=↑	root=↓	root=
			semi-empty spikes↑↑↑
	**C4 = KOH7 × D16**		
CKX expression 7 DAP	*CKX9*, *10*↓	*CKX9*, *10*↑	*CKX5*, *9*↑↑
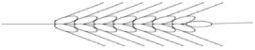	*CKX11*↑↑	*CKX11*↓↓	*CKX11*, *NAC2*↓↓
	*CKX2.1*, *2.2.2*↑	* CKX2.1 * , *2.2.2*↓	* CKX2.1 * , *2.2.2*↓
CKX expression root	*CKX3*, *8*↓	*CKX3*, *8*↑	* CKX8 * ↑↑
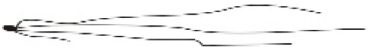	*CKX1*, *5*, *10*, *11*↑	*CKX1*, *5*, *10*, *11*↓	*CKX3*,*10*↓
	*NAC2*↑↑	*NAC2*↓↓	
yield-related traits	yield↓	yield↑	yield↑
	CKX act. spike=↓	CKX act. spike=↑	CKX act. spike=
	root↓	root↑	root=↓
	**C5 = D19 × D16**		
CKX expression 7 DAP	*CKX2.2.2*↑	* CKX2.2.2 * ↓	* CKX2.2.2 * ↓
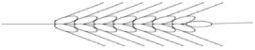	*CKX10*↓	*CKX10*↑	*CKX9*↑
CKX expression root	*CKX 5*, *8*, *NAC2*↑	*CKX5*, *8*, *NAC2*↓↓	* CKX3 * , *5*, *8*, *11*↓
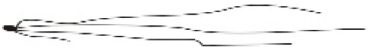	*CKX1*, *3*, *10*, *11*↑	*CKX1*, *3*, *10*, *11*↓	(*CKX10*, *NAC2*↑)
yield-related traits	yield↑	yield↓	yield↓↓
	CKX act. spike=↓	CKX act. spike=↑	CKX act. spike=↑
	root=	root=	root↓
	semi-empty spikes↓	semi-empty spikes↑	semi-empty spikes↑↑↑
	**C6 = D16 × D19**		
CKX expression 7 DAP	*CKX2.2.2*, *5*, *9*↓	*CKX2.2.2*, *5*, *9*↑	
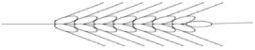	*CKX10*↑	* CKX10 * ↓	*CKX1*, *10*, *11*, *NAC2*↓
CKX expression root	*CKX5*, *8*, *NAC2*↓↓	*CKX5* , *8*, *NAC2*↑↑	* CKX1 * , *NAC2*↑↑
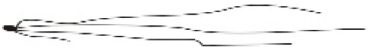	*CKX1*, *3*, *10*, *11*↓	*CKX1*, *3*,*10*, *11*↑	* CKX8 * , *10*↑
yield-related traits	yield↓	yield↑↑	yield↑
	CKX act. spike=↑	CKX act. spike=↓	CKX act. spike=
	root=	root=	root↓
	**C7 = D19 × KOH7**		
CKX expression 7 DAP	*CKX2.1*, *11*, *NAC2*↓	*CKX2.1*, *11*, *NAC2*↑	*CKX10*↑
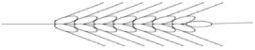	*CKX9*↑	* CKX9 * ↓	*CKX2.2.2*, *9*↓
			*NAC2*↓
CKX expression root	*NAC2*↓↓	*NAC2*↑↑	
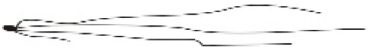	*CKX1*, *10*, *11*↓	*CKX1*, *10*, *11*↑	* CKX11 * ↑
	*CKX3*, *5*, *8*↑	* CKX3 * , *5*, *8*↓	*CKX3*, *5*, *8*, *10*↓
yield-related traits	yield↑	yield↓	yield↓
	CKX act. spike=	CKX act. spike=	CKX act. spike↑
	root=↑	root=↓	root=
		semi-empty spikes↑	semi-empty spikes↑↑
	**C8 = KOH7 × D19**		
CKX expression 7 DAP	*CKX5*, *9*, *10*↓	*CKX5* , *9*, *10*↑	* CKX9 * , *10*↑
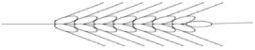	*CKX11*↑	* CKX11 * ↓	* CKX11 * , *NAC2*↓
	*CKX2.1*, *NAC2*=↑	* CKX2.1 * , *NAC2*=↓	*CKX2.1*, *2.2.2*↓
CKX expression root	*CKX8*↓↓	* CKX8 * ↑↑	* CKX3 * , *8*↑↑
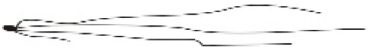	*CKX3*, *5*↓	*CKX3*, *5*↑	*CKX1*, *10*, *NAC2*↓
	*NAC2*↑↑	* NAC2 * ↓↓	
yield-related traits	yield↓	yield↑↑	yield↑↑
	CKX act. spike=	CKX act. spike=	CKX act. spike↑
	root=↓	root=↑	root=

Bold—cross number and parents.

**Table 2 ijms-24-08196-t002:** Coordinated expression of *TaCKX* GFMs and *NAC2* genes in 7 DAP spikes and seedling roots of the paternal parent (P) of four reciprocal crosses.

Cross	P Spike P Expr. +/P expr. −	P Root P Expr. +/P Expr. −	F_2_ Spike/Root (Yield, Root Mass in F_2_)
C1	*1*, *NAC2*/*5*, *9*	*3*, *5*, *NAC2*/*10*, *11*	*NAC2*+/*NAC2*+ y−, r = −, As−−
C2	*5*, *9*/*1*, *NAC2*	*10*, *11*/*3*, *5*, *NAC2*	*NAC2*−/*NAC2*− y = −, **r+**, As =
C3	*2.1*, *11*, *2.2.2*/*10*	*NAC2*, *1*, *5*, *10*, *11*/*3*	*10*−, *11*+/*10*+,*11*+, *NAC2*+ y−, r=, As−, s-e+++
C4	*9*, *10*/*2.1*, *11*, *2.2.2*	*3*, *8*/*1*, *5*, *10*, *11*, *NAC2*	*10*+, *11*−/*10*−, *11*−, *NAC2*− **y+, r = −**, As=
C5	*2.2.2*/*10*	?/*3*, *5*, *8*, *NAC2*, *1*, *10*, *11*	*10*−/*10*−, *NAC2*− y−, r−, As = +, s-e+++
C6	*2.2.2*, *5*, *9*/*10*	*3*, *5*, *8*, *NAC2*, *1*, *10*, *11*/none	*5*+, *10*−/ *5*+, *10*+, *NAC2*+ y+, r−, As=
C7	*2.1*, *11*, *NAC2*/*9*	*1*, *10*, *11*, *NAC2*/*3*, *5*, *8*	*11*+, *NAC2*+/*11*+, *NAC2*+ y−, r=, As+, s-e+++
C8	*5*, *9*, *10*/*2.1*, *11*, *NAC2*	*3*, *5*, *8*/*NAC2*	*5*+, *NAC2*−/*5*+, *NAC2*− **y++, r=**, As+

P—paternal parent; expr.+—upregulated; expr.—downregulated; 1, 3, 5, 9…-*TaCKX* GFMs, As—CKX activity spike; s-e+++—high number of semi-empty spikes, y—yield, r—root mass.

**Table 3 ijms-24-08196-t003:** Correlations between *TaCKX GFM* and *NAC2* expression in 7 DAP spikes or seedling roots, and yield-related traits in the group of maternal plants and F_2_, and paternal plants and F_2_ of reciprocal crosses.

		7 DAP Spike	Seedling Root	Yield-Related Traits	F_2_ Phenotype
		**Seed Number**			
**C1** **S12B × S6C**	M + F_2_	*CKX1*−, ***CKX5*−**	***CKX11*+**	spike number+	yield−−, CKX act. −−, root=−, CKX act. root++
	P + F_2_	nc	*CKX11*+	**spike number+**	
**C2** **S6C × S12B**	M + F_2_	nc	nc	CKX act. root−, **spike number++**	yield−, CKX act.=, root+, CKX act. root=
	P + F_2_	nc	*CKX3*+	CKX act. root−, **spike number+**	
**C3** **D16 × KOH7**	M + F_2_	***CKX2.1*+,***CKX2.2.2*+	***CKX3*+, *CKX8*+,***CKX11*−	**plant height+, spike number+**	yield−, CKX act. −−, root=−, semi-empty spikes++
	P + F_2_	***CKX2.1*+, *CKX2.2.2*+**	*CKX1*−, ***CKX3*****++, *CKX8*+, *CKX11*−**	**plant height+, spike number+**	
**C4** **KOH7 × D16**	M + F_2_	nc	nc	**plant height+, spike number+**	yield+, CKX act.=, root=
	P + F_2_	nc	nc	plant height+, **spike number+**	
**C5** **D19 × D16**	M + F_2_	***CKX1*−**	***CKX3*+, *CKX5*+, *NAC2*+**	**plant height+, spike number+**	yield−−, CKX act.+, root=−, semi-empty spikes++
	P + F_2_	nc	***CKX3*+,** *NAC2*+	plant height+ **spike number+**	
**C6** **D16 × D19**	M + F_2_	nc	nc	**spike number+**	yield+, CKX act.=, root=−
	P + F_2_	nc	nc	**spike number+**	
**C7** **D19 × KOH7**	M + F_2_	nc	*CKX1*−	**spike number++**	yield−, CKX act.+, root=−, semi-empty spikes++
	P + F_2_	nc	*CKX1*−	**spike number++**	
**C8** **KOH7 × D19**	M + F_2_	nc	***CKX1*−**	empty spikes−, **spike number+**	yield++, CKX act.=+, root=
	P + F_2_	***CKX2.1*+**	***CKX1*−**	**empty spikes−, spike number+**	
		**Seed yield**			
**C1** **S12B × S6C**	M + F_2_	*CKX11*−, *NAC2*−	***CKX11*+**	**spike number+, seed number++**	yield−, CKX act. −−, root=−, CKX act. root++
	P + F_2_	nc	*CKX11*+	**spike number+, seed number++**	
**C2** **S6C × S12B**	M + F_2_	nc	*CKX10*+	**spike number+, seed number++**	yield−, CKX act.=, root+, CKX act. root=
	P + F_2_	nc	*CKX10*+, *CKX3*+	**spike number+, seed number++**	
**C3** **D16 × KOH7**	M + F_2_	***CKX2.1*+, *CKX2.2.2*+**	***CKX3*+, *CKX8*+**	plant height+, **spike number+, seed number++**	yield−, CKX act. −−, root=−, semi-empty spikes++
	P + F_2_	*CKX2.1*+, *CKX2.2.2*+	***CKX3*+**, *CKX8*+	**plant height+, spike number+, seed number++**	
**C4** **KOH7 × D16**	M + F_2_	nc	nc	**plant height+, spike number+, seed number++**	yield+, CKX act.=, root=
	P + F_2_	nc	nc	**plant height+, spike number+, seed number++**	
**C5** **D19 × D16**	M + F_2_	***CKX1*−**	***CKX3*+, *CKX5*+, *NAC2*+**	**plant height+, spike number+, seed number++**	yield−−, CKX act.+, root=−, semi-empty spikes++
	P + F_2_	nc	***CKX3*+, *NAC2*+**	plant height+, **spike number+, seed number++**	
**C6** **D16 × D19**	M + F_2_	***CKX2.1*+**	nc	**plant height+, spike number+, seed number++**	yield+, CKX act.=, root=−
	P + F_2_	***CKX1*−**	nc	**plant height+, spike number+, seed number++**	
**C7** **D19 × KOH7**	M + F_2_	nc	nc	**plant height+, spike number++, seed number++**	yield−, CKX act.+, root=−, semi-empty spikes++
	P + F_2_	nc	nc	**plant height+, spike number++, seed number++**	
**C8** **KOH7 × D19**	M + F_2_	nc	***CKX1*−**	**spike number+, seed number++**	yield++, CKX act.=+, root=
	P + F_2_	nc	***CKX1*−**	**spike number+, seed number++**	
		**Spike number**			
**C1** **S12B × S6C**	M + F_2_	*CKX2.1*+, ***CKX2.2.2*+, *CKX11*−**	***CKX11*+**	plant height+	yield−, CKX act.−−, root=−, CKX act. root++
	P + F_2_	***CKX2.1*+, CKX*2.2.2*+**	nc	**CKX act.+, plant height+**	
**C2** **S6C × S12B**	M + F_2_	***CKX2.2.2*+**	nc	nc	yield−, CKX act.=, root+, CKX act. root=
	P + F_2_	nc	*NAC2*−	nc	
**C3** **D16 × KOH7**	M + F_2_	nc	*CKX8*+	nc	yield−, CKX act. −−, root=−, semi-empty spikes++
	P + F_2_	nc	***CKX8*+**	nc	
**C4** **KOH7 × D16**	M + F_2_	nc	***CKX5*+**	nc	yield+, CKX act.=, root=
	P + F_2_	nc	***CKX5*+**	nc	
**C5** **D19 × D16**	M + F_2_	nc	*CKX1*−, *CKX5*+, *NAC2*+	nc	yield−−, CKX act.+, root=−, semi-empty spikes++
	P + F_2_	nc	***CKX1*−**, *CKX5*+, *NAC2*+	nc	
**C6** **D16 × D19**	M + F_2_	nc	nc	nc	yield+, CKX act.=, root=−
	P + F_2_	***CKX2.1*+**	nc	nc	
**C7** **D19 × KOH7**	M + F_2_	nc	*CKX1*−	nc	yield−, CKX act.+, root=−, semi-empty spikes++
	P + F_2_	nc	*CKX1*−, ***CKX8*****−**	nc	
**C8** **KOH7 × D19**	M + F_2_	nc	***CKX1*−, *CKX5*−, *CKX8*−**	nc	yield++, CKX act.=+, root=
	P + F_2_	nc	***CKX1*−, *CKX5*−**	nc	
		**TGW**			
**C1** **S12B × S6C**	M + F_2_	***CKX2.1*+, *CKX10*+, *NAC2*+**	nc	nc	yield−, CKX act. −−, root=−, CKX act. root++
	P + F_2_	***CKX2.2.2*+**, *NA2C*+	nc	nc	
**C2** **S6C × S12B**	M + F_2_	nc	nc	**CKX act. root++,** seed number−	yield−, CKX act.=, root+, CKX act. root=
	P + F_2_	nc	nc	**CKX act. root++, seed number−**	
**C3** **D16 × KOH7**	M + F_2_	nc	nc	**plant height+, yield+, semi-empty−** **, seed number+**	yield−, CKX act. −−, root=−, semi-empty spikes++
	P + F_2_	nc	nc	**plant height+, yield+**	
**C4** **KOH7 × D16**	M + F_2_	***CKX2.2.2*−**	** *CKX10* ** **+, *NAC2*+**	**CKX act.−**	yield+, CKX act.=, root=
	P + F_2_	***CKX2.2.2*−**	***CKX10*+,***NAC2*+	CKX act.−	
**C5** **D19 × D16**	M + F_2_	nc	*NAC2*+	**yield+, seed number+**	yield−−, CKX act.+, root=−, semi-empty spikes++
	P + F_2_	nc	*NAC2*+	**yield+**	
**C6** **D16 × D19**	M + F_2_	nc	nc	plant height+	yield+, CKX act.=, root=−
	P + F_2_	nc	nc	**plant height+**	
**C7** **D19 × KOH7**	M + F_2_	***CKX2.1*−, *CKX2.2.2*−, *CKX9*−**	nc	**spike length−**	yield−, CKX act.+, root=−, semi-empty spikes++
	P + F_2_	***CKX2.1*−,***CKX2.2.2*−, ***CKX5*+**	nc	nc	
**C8** **KOH7 × D19**	M + F_2_	*CKX2.1*−, ***CKX2.2.2*−**	nc	nc	yield++, CKX act.=+, root=
	P + F_2_	***CKX2.1*−, *CKX2.2.2*−!, *CKX10*−**	nc	nc	
		**Root mass**			
**C1** **S12B × S6C**	M + F_2_	***CKX5*+,***CKX11*+**, *NAC2*+**	*CKX1*−, *CKX3*−, *CKX10*−, *CKX11*−,	seed number−	yield−, CKX act. −−, root=−, CKX act. root++
	P + F_2_	nc	***NAC2*−**, *CKX1*−, *CKX10*−, ***CKX11*****−**	seed number−	
**C2** **S6C × S12B**	M + F_2_	*NAC2*−	nc	nc	yield−, CKX act.=, root+, CKX act. root=
	P + F_2_	nc	nc	nc	
**C3** **D16 × KOH7**	M + F_2_	***CKX1*+**, *CKX11*+	** *CKX5* ** **+**	nc	yield−, CKX act.−−, root=−, semi-empty spikes++
	P + F_2_	*CKX1*+, ***CKX11*+, *NAC2*+**	** *CKX5* ** **+, *CKX8*−**	nc	
**C4** **KOH7 × D16**	M + F_2_	nc	***CKX3*+**, *CKX8*+, *NAC2*−	**CKX act.+**	yield+, CKX act.=, root=
	P + F_2_	***CKX2.1*+, *CKX2.2.2*+, *CKX10*+**	** *CKX3* ** **+, *CKX8*+, *NAC2*−**	nc	
**C5** **D19 × D16**	M + F_2_	***NAC2*−**	***CKX1*−**, *CKX11*+	nc	yield−−, CKX act.+, root=−, semi-empty spikes++
	P + F_2_	***CKX2.1*+**	nc	nc	
**C6** **D16 × D19**	M + F_2_	***CKX2.1*+, *CKX2.2.2*+**	nc	**yield+**	yield+, CKX act.=, root=−
	P + F_2_	***CKX2.2.2*+**	nc	**plant height+**	
**C7** **D19 × KOH7**	M + F_2_	nc	** *CKX1* ** **−**	nc	yield−, CKX act.+, root=−, semi-empty spikes++
	P + F_2_	** *NAC2* ** **−**	** *CKX1* ** **−**	**spike length+**	
**C8** **KOH7 × D19**	M + F_2_	** *CKX10* ** **−**	** *CKX3* ** **−**	nc	yield++, CKX act.=+, root=
	P + F_2_	nc	** *CKX3* ** **−**	**yield+**	
		**Plant height**			
**C1** **S12B × S6C**	M + F_2_	nc	nc	nc	yield−, CKX act. −−, root=−, CKX act.root++
	P + F_2_	nc	nc	nc	
**C2** **S6C × S12B**	M + F_2_	***CKX11*+**	** *CKX1* ** **−**	nc	yield−, CKX act.=, root+, CKX act.root=
	P + F_2_	nc	*CKX5*−, *CKX11*+	nc	
**C3** **D16 × KOH7**	M + F_2_	nc	*CKX11*−	nc	yield−, CKX act.−−, root=−, semi-empty spikes++
	P + F_2_	nc	*CKX11*−	nc	
**C4** **KOH7 × D16**	M + F_2_	nc	nc	nc	yield+, CKX act.=, root=
	P + F_2_	***CKX2.2.2*−**	** *NAC2* ** **+**	nc	
**C5** **D19 × D16**	M + F_2_	*CKX1*−	*CKX1*−, *NAC2*+	nc	yield−−, CKX act.+, root=−, semi-empty spikes++
	P + F_2_	nc	*CKX1*−, ***NAC2*****+**	nc	
**C6** **D16 × D19**	M + F_2_	***CKX1*−, *CKX5*−**	***CKX5*−**	nc	yield+, CKX act.=, root=−
	P + F_2_	***CKX5*−**	*CKX5*−	nc	
**C7** **D19 × KOH7**	M + F_2_	nc	nc	nc	yield−, CKX act.+, root=−, semi-empty spikes++
	P + F_2_	nc	nc	nc	
**C8** **KOH7 × D19**	M + F_2_	***CKX2.2.2*−**, *CKX9*−	*CKX11*+	nc	yield++, CKX act.=+, root=
	P + F_2_	***CKX9*−**	nc	nc	
		**Spike length**			
**C1** **S12B × S6C**	M + F_2_	***NAC2*−**	nc	semi-empty spikes−, seed number+, **yield+**	yield−, CKX act.−−, root=−, CKX act. root++
	P + F_2_	*NAC2*−	nc	**spike number+, seed number+,** yield+	
**C2** **S6C × S12B**	M + F_2_	***CKX9*+**, *CKX11*+	nc	**plant height+**	yield−, CKX act.=, root+, CKX act. root=
	P + F_2_	nc	nc	nc	
**C3** **D16 × KOH7**	M + F_2_	*CKX2.1*+	** *CKX3* ** **+, *CKX11*−**	**plant height+, seed number+,** yield+	yield−, CKX act.−−, root=−, semi-empty spikes++
	P + F_2_	***CKX2.1*+, *CKX2.2.2*+**	*CKX1*−, ***CKX3*++, *CKX11*−**	**plant height+, seed number+, yield+**	
**C4** **KOH7 × D16**	M + F_2_	***CKX1*+, *CK*X10+**	nc	**plant height+, seed number+,** yield+	yield+, CKX act.=, root=
	P + F_2_	nc	nc	**seed number+, yield+**	
**C5** **D19 × D16**	M + F_2_	***CKX5*−, *CKX11*+, *NAC2*−**	** *CKX11+* **	nc	yield−−, CKX act.+, root=−, semi-empty spikes++
	P + F_2_	nc	nc	seed number+, yield+	
**C6** **D16 × D19**	M + F_2_	nc	nc	**spike number+, seed number+, yield+**	yield+, CKX act.=, root=−
	P + F_2_	nc	nc	**seed number+, yield+**	
**C7** **D19 × KOH7**	M + F_2_	***CKX1*+**, *CKX9*+	nc	**seed number+**	yield−, CKX act.+, root=−, semi-empty spikes++
	P + F_2_	*CKX9*+, ***NAC2*−**	nc	nc	
**C8** **KOH7 × D19**	M + F_2_	nc	nc	nc	yield++, CKX act.=+, root=
	P + F_2_	***CKX2.1*+**	nc	**seed number+**	
		**Semi-empty spikes**			
**C1** **S12B × S6C**	M + F_2_	*CKX5*+	***CKX5*−**, *CKX10*−	nc	yield−, CKX act. −−, root=−, CKX act. root++
	P + F_2_	***CKX5*+!**, *CKX9*+, ***CKX11*+**	nc	nc	
**C2** **S6C × S12B**	M + F_2_	***CKX9*+**	nc	nc	yield−, CKX act.=, root+, CKX act. root=
	P + F_2_	*CKX9*+	** *CKX8* ** **−**	empty spikes+	
**C3** **D16 × KOH7**	M + F_2_	***CKX9*+**	nc	nc	yield−, CKX act.−−, root=−, semi-empty spikes++
	P + F_2_	***CKX11*−**	nc	nc	
**C4** **KOH7 × D16**	M + F_2_	nc	nc	**spike number+**	yield+, CKX act.=, root=
	P + F_2_	nc	nc	**spike number+**	
**C5** **D19 × D16**	M + F_2_	nc	nc	nc	yield−−, CKX act.+, root=−, semi-empty spikes++
	P + F_2_	***CKX2.1*−**	nc	nc	
**C6** **D16 × D19**	M + F_2_	*CKX5*+	** *CKX5* ** **+**	**spike number+**	yield+, CKX act.=, root=−
	P + F_2_	*CKX5*+	nc	**seed number+**	
**C7** **D19 × KOH7**	M + F_2_	***CKX10*+**	nc	nc	yield−, CKX act.+, root=−, semi-empty spikes++
	P + F_2_	nc	nc	nc	
**C8** **KOH7 × D19**	M + F_2_	nc	***CKX5*−*, CKX8*−,***NAC2*−	nc	yield++, CKX act.=+, root=
	P + F_2_	***CKX10*+**	*CKX5*−, ***NAC2*−**	nc	

All correlation coefficients ≥0.60; bold—significant correlation coefficients; nc—no correlation; +—positive correlation; =+—low positive correlation; ++—strong positive correlation; −—negative correlation; =−—low negative correlation; −−—strong negative correlation.

## Data Availability

All data generated or analysed during this study are included in this published article [and its Appendix A files].

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
