# Peer review of "Transgenerational Paternal Inheritance of TaCKX GFMs Expression Patterns Indicate a Way to Select Wheat Lines with Better Parameters for Yield-Related Traits"

_ijms, 2023, doi:10.3390/ijms24098196_

Round 1

Reviewer 1 Report

The work by Szalaet al. explored the expression of TaCKX GFMs and TaNAC2-5A as candidates for yield selection in wheat. This is an accurate and pertinent work, which addresses the expression diversity in a model crop species. Overall, the work is well written, statistically up to date, and highlights key findings. However, before commending acceptance, I have the following suggestions.

First, the paragraph of the introduction where goals are described (paragraph in L128 must go first) should also make explicit the research gap, the research hypothesis and the expected results (rather than the actual results from L131 after “unexpectedly”). This will allow readers focusing on explicit expectations when approaching the report.

Second, authors should mention in the introduction the broad consequences of TaCKX GFMs and TaNAC2-5A genes for other pleiotropic traits. For instance, are there known yield trade-off of these genes under abiotic stresses responses. Aslo, I am missing key references regarding the polygenetic nature of the genomic architecture for the linked traits that may not be all Mendelian (contrary to L69). For instance, other gene families have also been linked with acceptable yield index  under abiotic stresses, drought in particular, as indicated in: (i) Plant Science 2016 242:250 for the ERECTA gene family in association with the AP2 domain, (ii) BMC Genetics 2012 13:58 for the ASR family f in association with the ABA-dependent MYB, and (iii) Theor Appl Genet 2012 125(5):1069-85 for the DREB transcription factor family pleotropic in several pathways with the WRKY transcription factor. Authors should discussed these cases explicitly and make a clear point since the introduction on why focusing only on the TaCKX GFMs and TaNAC2-5A genes seems biased. Please revisit this point at the end of the discussion and recommend expanding the analysis to other gene families.

Third, my major analytical suggestion is to leave out colors and highlights from the main three tables and the third figure. Besides, every time a barplot is presented (figures 1) please include significance as error bars on top of the individual bars, do not use guiding horizontal bars and optimize the color selection to improve readability.

Fourth, please also complement the tissue-dependent expression with explicit gene-environment associations, which may have higher power for the detection of context-dependent environmental responses (refer to the seminal review Front Genet 2022 13:910386) at the transcriptomic level. The gene-environment approach has been validated for (refer to Front Genet 2019 10:954, Front Plant Sci 2018 9:128 and Genes 2021 12:556), yet authors should explicitly comment on the significance of the replication level utilized to gather the expression profiles, for which a preliminary power analysis would be insightful.

As closure, please include a perspective section at the end of the discussion in L546 with recommendation on how to better integrate omic technologies with modern analytical approaches to assess gene expression for yield traits in crop species by referring to the seminal reviews Front Plant Sci 2020 11:583323 and Front Genet 2020 11:564515.

Finally, in terms of writing, the abstract is excessively synthetic and does not follow the ABT recommendation (see this card: https://entomologychallenges.files.wordpress.com/2018/10/abt-shorthand-reference-card.pdf) for abstracts (in L11). Please improve it by including a research gap.

Author Response

Dear Reviewer 1,

Thank you for your effort to improve our manuscript. We try to address most of your comments and suggestions in our revised manuscript. Please find our responses to each of your points below.

First, the paragraph of the introduction where goals are described (paragraph in L128 must go first) should also make explicit the research gap, the research hypothesis and the expected results (rather than the actual results from L131 after “unexpectedly”). This will allow readers focusing on explicit expectations when approaching the report.

Our response: Thank you for this reasonable suggestion. The paragraph was corrected.

Second, authors should mention in the introduction the broad consequences of TaCKX GFMs and TaNAC2-5A genes for other pleiotropic traits. For instance, are there known yield trade-off of these genes under abiotic stresses responses.

Our response: Thank you. We mention the pleiotropic traits regulated by these genes.

Aslo, I am missing key references regarding the polygenetic nature of the genomic architecture for the linked traits that may not be all Mendelian (contrary to L69). For instance, other gene families have also been linked with acceptable yield index  under abiotic stresses, drought in particular, as indicated in: (i) Plant Science 2016 242:250 for the ERECTA gene family in association with the AP2 domain, (ii) BMC Genetics 2012 13:58 for the ASR family f in association with the ABA-dependent MYB, and (iii) Theor Appl Genet 2012 125 (5):1069-85 for the DREB transcription factor family pleotropic in several pathways with the WRKY transcription factor. Authors should discussed these cases explicitly and make a clear point since the introduction on why focusing only on the TaCKX GFMs and TaNAC2-5A genes seems biased. Please revisit this point at the end of the discussion and recommend expanding the analysis to other gene families.

Our response: Thank you for this point. I included short information about the ‘polygenetic nature of genomic architecture…’ and suggested two more review articles. I do not think that it needs to be discussed since these research is not closely related to ours (explained below).

Third, my major analytical suggestion is to leave out colors and highlights from the main three tables and the third figure. Besides, every time a barplot is presented (figures 1) please include significance as error bars on top of the individual bars, do not use guiding horizontal bars and optimize the color selection to improve readability.

Our response: Thank you for suggestions. We removed highlights from Table 3, but prefer to leave the colors and highlights in Table 1 and 2 as well as in Fig. 3. They clearly indicate described results and are included in the text as well.

Fig. 1 and Fig. S1 – we added significance (to mother and to pater) on the top of each bar as suggested and improved visibility.

Fourth, please also complement the tissue-dependent expression with explicit gene-environment associations, which may have higher power for the detection of context-dependent environmental responses (refer to the seminal review Front Genet 2022 13:910386) at the transcriptomic level. The gene-environment approach has been validated for (refer to Front Genet 2019 10:954, Front Plant Sci 2018 9:128 and Genes 2021 12:556), yet authors should explicitly comment on the significance of the replication level utilized to gather the expression profiles, for which a preliminary power analysis would be insightful.

Our response: I collect all your articles suggested (8) to include in our manuscript and try to find any relation to our research, but unfortunately, without any success. Please note that our research was carried out in a controlled environment (in the growth chamber) to test the inheritance of expression patterns for selected yield-related genes. There is no point in discussing gene-environment interactions. I will remember your articles, when we move our research to environment.

As closure, please include a perspective section at the end of the discussion in L546 with recommendation on how to better integrate omic technologies with modern analytical approaches to assess gene expression for yield traits in crop species by referring to the seminal reviews Front Plant Sci 2020 11:583323 and Front Genet 2020 11:564515.

Our response: Sorry, but based on our results there is not appropriate to discuss “on how to better integrate omic technologies with modern analytical approaches…’ The reviews you suggested are not relevant to our research, which were carried out under laboratory conditions (not environment), on segregating F2 generation of self-pollinating cereal species.

Finally, in terms of writing, the abstract is excessively synthetic and does not follow the ABT recommendation (see this card: https://entomologychallenges.files.wordpress.com/2018/10/abt-shorthand-reference-card.pdf) for abstracts (in L11). Please improve it by including a research gap.

Our response: The abstract is already too long according to MDPI requirements. Since other reviewers have no comments, the authors would like to leave the abstract as it is.

Thank you and best regards,                                                                                                 

Anna Nadolska-Orczyk

Reviewer 2 Report

I have the following questions/suggestions before taking a final decision:

  1. What is the function of TaCKX genes in wheat plants, and how do they regulate yield-related traits?
  2. Line 14: “The traits were tested in 7 DAP spikes and seedling roots of maternal” Similarly TGW, please add full form at first appearance then use abbreviations. Please check the whole manuscript.
  3. Figure 1: Some bars have SD/SE values but some don’t have, please recheck.
  4. Each reaction was carried out in 3 biological and 3 technical replicates”. Normally Biological replicates are used, then why the authors used technical replicates?
  5. What is the role of TaNAC2-5A in the regulation of yield-related traits, and how is its expression inherited in wheat plants?
  6. What is the significance of the cooperative or opposite functions of some pairs or groups of genes in regulating yield-related traits in wheat plants?
  7. Which specific TaCKX genes were found to be correlated with individual yield-related traits, and how?
  8. What is the significance of the newly shown data of non-Mendelian epigenetic inheritance in crossing strategies to obtain a high-yielding F2 generation?
  9. What are the potential applications of this knowledge in speeding up breeding processes in wheat plants?

English is fine please check it again for clarification.

Author Response

Dear Reviewer 2,

Thank you for your questions and corrections that were helpful for improving the manuscript. Some of them are addressed in this new version of the manuscript, and part of them are explained below the questions.

  1. What is the function of TaCKX genes in wheat plants, and how do they regulate yield-related traits?

Our response: Thank you. I included the basic information in the introduction and they are widely discussed in the discussion.

  1. Line 14: “The traits were tested in 7 DAP spikes and seedling roots of maternal” Similarly TGW, please add full form at first appearance then use abbreviations. Please check the whole manuscript.

Our response: Thank you. It is done.

  1. Figure 1: Some bars have SD/SE values but some don’t have, please recheck.

Our response: The F2 generation of tested crosses is a segregating population, therefore, phenotypic traits were measured for six individual F2 plants (information included).

  1. “Each reaction was carried out in 3 biological and 3 technical replicates”. Normally Biological replicates are used, then why the authors used technical replicates?

Our response: We used technical replicates to check the accuracy between separate reactions.

  1. What is the role of TaNAC2-5A in the regulation of yield-related traits, and how is its expression inherited in wheat plants?

Our response: This information are included in the introduction, results, and discussion.

  1. What is the significance of the cooperative or opposite functions of some pairs or groups of genes in regulating yield-related traits in wheat plants?

Our response: This is a very general question and I do not know what do you thing about, besides this, which is already discussed (in our long discussion). Their main role is to maintain phytohormonal homeostasis and through this to regulate the development of individual organs.

  1. Which specific TaCKX genes were found to be correlated with individual yield-related traits, and how?

Our response: I am again confused with this question. The answer is in the results and discussed in the discussion.

  1. What is the significance of the newly shown data of non-Mendelian epigenetic inheritance in crossing strategies to obtain a high-yielding F2 generation?

Our response: This has already been explained in the manuscript.

  1. What are the potential applications of this knowledge in speeding up breeding processes in wheat plants?

Our response: It will be possible to test the expression pattern of selected TaCKX GFMs and TaNAC2-5A in the components for crossing and decide which is the parent component and which is the mother component (included in the manuscript).

Thank you and best regards,

Anna Nadolska-Orczyk

Round 2

Reviewer 1 Report

Thanks to the authors for their careful revision and rebuttal letter, with which I agree. I still think that text highlight with colours within tables 1, 2 and figure 3 is far unconventional and could be substitute by a rather more simple bold text. Yet I leave this detail to the production office. Similarly, readability of figure 1 could improve if authors did not use horizontal grey guiding lines, another suggestion worth considering at the production stage. 

Reviewer 2 Report

All my concern addressed properly. 

Language is fine